# Survey of Multimodal Medical Question Answering

**Hilmi Demirhan [1,\*] and Wlodek Zadrozny [2]**

[1] Congdon School of Supply Chain, Business Analytics and Information Systems, University of North Carolina Wilmington, Wilmington, NC 28403, USA

[2] Department of Computer Science, University of North Carolina Charlotte, Charlotte, NC 28223, USA; wzadrozn@charlotte.edu

[\*] Correspondence: demirhanh@uncw.edu

**Abstract:** Multimodal medical question answering (MMQA) is a vital area bridging healthcare and Artificial Intelligence (AI). This survey methodically examines the MMQA research published in recent years. We collect academic literature through Google Scholar, applying bibliometric analysis to the publications and datasets used in these studies. Our analysis uncovers the increasing interest in MMQA over time, with diverse domains such as natural language processing, computer vision, and large language models contributing to the research. The AI methods used in multimodal question answering in the medical domain are a prominent focus, accompanied by applicability of MMQA to the medical field. MMQA in the medical field has its unique challenges due to the sensitive nature of medicine as a science dealing with human health. The survey reveals MMQA research to be in an exploratory stage, discussing different methods, datasets, and potential business models. Future research is expected to focus on application development by big tech companies, such as MedPalm. The survey aims to provide insights into the current state of multimodal medical question answering, highlighting the growing interest from academia and industry. The identified research gaps and trends will guide future investigations and encourage collaborative efforts to advance this transformative field.

**Keywords:** multimodal medical question answering; natural language processing; visual question answering; medical question answering; survey; artificial intelligence; computer vision; large language models; medical science; health; telehealth; telemedicine; sensitive data; research gaps; AI methods

## 1. Introduction

Multimodal Medical Question Answering (MMQA) systems have gained prominence due to several compelling factors. There are a lot of different data in healthcare today, like medical records, images, audio, and videos. MMQA systems help by putting this diverse data together, which is a crucial need. Healthcare questions can be quite complex, requiring an approach that MMQA excels at—deciphering and merging insights from various data sources.

Advanced Artificial Intelligence (AI), especially in areas like Natural Language Processing (NLP) and Computer Vision, empowers MMQA. These AI systems can handle complex textual and visual data effectively. Clinical decision support is on the rise because there are not always enough healthcare professionals. MMQA can alleviate their workload, reduce errors, and improve efficiency.

In healthcare, quick and precise diagnostics are critical. MMQA systems, by aligning textual descriptions with medical imagery, offer faster and more accurate diagnosis. The rise of personalized medicine has underscored the importance of integrating diverse patient data, and MMQA systems are particularly skilled at providing personalized treatment recommendations. Technology advancements and the availability of high-quality multimodal datasets have further driven MMQA research, making it even more relevant.

The healthcare sector, along with AI researchers and technologists, is actively promoting MMQA systems' development and deployment. These systems have the potential to enhance clinical decision support, diagnostic accuracy, and personalized healthcare, which, in turn, can improve overall healthcare efficiency and outcomes.

Interpreting medical text and visual information requires expertise in the medical domain. Healthcare professionals, such as doctors and nurses go through extensive training in medical knowledge. They are equipped to understand and interpret medical text and visual information. This includes good understanding of medical terminology, understanding the various systems of the human body, and being able to interpret diagnostic tests. It also involves the ability to synthesize information from multiple sources to make informed decisions about patient care. In addition to formal medical training, staying up to date with current research and best practices is important for healthcare professionals in order to provide the best care for their patients.

Medical decision support techniques like Fuzzy Cognitive Maps(FCMs) can be used for modeling complex systems [1] . It can represent the relationships between various factors that may impact a particular decision, such as patient symptoms, diagnostic test results, and treatment options. A shortage of medical professionals can lead to a higher workload for those who are available. This can increase the risk of errors due to working long hours, lack of sleep, or high levels of stress. It is important for medical practitioners to prioritize self-care and take steps to manage their workload in order to reduce the risk of errors by utilizing technology to help streamline tasks and reduce the burden. ChatGPT and other large language models are also being used for medical question answering [2] to help patients and clinicians with their first triage.

Natural language processing, as an aspect of artificial intelligence, focuses on increasing the ability of natural language communication of computers to the levels of human intelligence. Understanding the meaning of sentences, sentiment analysis, speech recognition, and machine translation are some of the common tasks of natural language processing. Computers can be used for retrieving information from large texts and databases. The text retrieval conference that was started in 1993 was designed for encouraging information retrieval from large-scale datasets [3]. The first Biomedical Semantic Indexing and Question Answering (BioASQ) challenge started in 2013. It includes various tasks such as hierarchical text classification, information retrieval and text quality assurance [4].

Computer vision, as part of artificial intelligence, focuses on interpreting and analyzing visual data similar to how humans do. The common tasks of computer vision include recognizing objects, image segmentation, and understanding context and relationships from images and videos. Visual Question Answering (VQA) can be used for image and video based information retrieval [5,6]. In recent years the need for multimodal question answering systems that are using machine learning and deep learning algorithms has increased.

Question answering systems are trained to return the most relevant information from specialized databases or other data resources. Development of advanced AI algorithms for clinical decision support can improve the efficiency of multimodal medical information systems that can find answers to questions of natural language. In recent years, availability of various large-scale text, image, and video based datasets has contributed efficient algorithm development to understand and interpret medical videos. The first Medical Video Question Answering Challenge (MedVidQA) organized by the National Institute of Health in 2022 [7] focused mainly on the real-world scenario of finding the relevant part in medical videos.

*1.1. Research Questions*

This work provides a review of the literature on the current state of the art of the problem. It will also present the techniques used for multimodal medical question answering. The research questions of this survey are listed below:

- What capabilities and features are offered by MMQA tools and research methods?

- What are the available datasets and tools in the commercial and open source space, and can the datasets be classified according to size?
- How have recent advances in AI influenced the development and application of medical question answering systems, and what are the key challenges in this evolving field?

### 1.2. Methodology

A search on Google Scholar was performed with the keywords "Multimodal Medical Question Answering" to find the methods, tools, and datasets used in the MMQA field. The authors of this survey paper, Hilmi Demirhan and his PhD advisor, Prof. Dr. Wlodek Zadrozny, were involved in the paper selection process. A total of 51 articles were selected to be analyzed for the methods and datasets they use. The papers were manually selected by reading the titles and abstracts to check for the relevance of the paper to the MMQA field. Non-English publications, books, theses, irrelevant document types, and completely off-topic papers were eliminated. The language was limited to English, and some of the retrieved documents were eliminated to meet this criterion. Articles that were published before 2017 were also eliminated.

All four search terms "multimodal, medical, question, answering" did not appear in every resulting publication, the topic of numerous papers being not related. In several cases, only the terms "question answering" appeared. All selected 51 papers concerned visual+text question answering. Medical Question Answering papers which did not include visual queries were eliminated as well. A total of 49 papers were published after 2017, and two papers were published in 2017. Apart from those cases, the subject of a few publications were found to be irrelevant to the search terms. Since answering medical questions is the main topic of the survey; documents that were not related to medically related fields were eliminated as well. The search terms were checked in the title, abstract, and keyword sections of the publications. Synonym words were not searched in this assessment because Google Scholar was very efficient in retrieving related papers. After preprocessing, documents were quickly scanned to see if they had built an applied model for multimodal medical question answering. Theoretical papers were eliminated as well. Papers without full text were also excluded.

### 1.3. Motivation

The need for continuous improvement of medical decision support systems for accurate analysis and predictions has increased due to worker shortages and disparity gaps in healthcare. A Multimodal Medical Question Answering system can provide better and more accurate answers by using multiple modalities. In the medical field, providing fast and effective care is critical. A Multimodal framework will improve the speed of answering questions. Assessing a health condition based on a text based description can be hard. For example, doctors will understand heart conditions better if they perform image and video based cardiovascular scanning.

Multimodal medical question answering can help to make accurate clinical decisions in healthcare. With the increasing availability of electronic medical records, medical imaging, and other forms of medical data, there is a growing need for automated systems that can help clinicians make accurate diagnoses.

Multimodal question answering systems have the potential to provide a more comprehensive approach to clinical decision-making by integrating multiple forms of data, such as medical images, text, and other sensor data. By combining these different modalities, multimodal question answering systems can provide a more holistic view of patient data. They can also allow for more accurate and personalized treatment recommendations.

### 1.4. Organization

The paper follows a structure to present its research on multimodal medical question answering. It begins with an introduction that highlights the need for efficient clinical

decision support systems in healthcare. Section 2 reviews previous research in the field of multimodal medical question answering (QA) using natural language processing and computer vision techniques. Section 3 lists the available multimodal QA datasets. Section 4 outlines the research framework, including algorithms and evaluation metrics used to implement and test multimodal medical question answering systems. Section 5 explores the current state-of-the-art large language models and their significance in healthcare question answering. Sections 6 and 7 present potential methodology and preliminary outcomes from the implementation and testing of the system. The paper also touches on industrial research and startup investment opportunities in the field in Section 6, and concludes with challenges, a summary of key points, and the importance of efficient clinical question-answer systems in Sections 7 and 8. The reference section lists all sources cited throughout the paper.

## 2. Multimodal Medical Question Answering

A Multimodal Medical Question Answering system uses natural language processing and computer vision algorithms to provide answers to medical questions. The system needs to be trained on medical datasets that can contain text, audio, image, and video. After training the system, it can provide answers to new unseen medical questions. This can be done analyzing the texts from doctor's notes, visual scan data such as X-Ray images, and video recordings of the patient. It can help healthcare providers diagnose and treat patients by answering questions about specific medical conditions, treatments, and procedures.

There are several approaches to building multimodal question answering systems. A multimodal fusion approach involves combining the outputs of multiple modality-specific models to answer a question [8]. For example, a model might use the output of a text-based question answering model and an image-based object recognition model to answer a question about an image. The multitask learning approach involves training a single model to perform multiple tasks, such as answering questions and recognizing images [9]. The model can then use the information it has learned from one task to help perform the other task. The transfer learning approach involves pre-training a model on a large dataset and then fine-tuning it for a specific task [10]. This can be an effective approach for multimodal question answering, as it allows the model to leverage the knowledge it has learned from the pretraining dataset to better understand the task at hand. The combination approach of information from multiple modalities involves using information from different modalities (e.g., text, images, video) to answer a question. For example, a model might use both the text of a question and an image to answer a question about the image.

### 2.1. Text Analysis and Understanding

Question answering systems are a subfield of text analysis and understanding, which is a subfield of natural language processing (NLP). Text analysis and understanding focuses on developing algorithms and techniques to automatically extract information and interpret the meaning of text data. This can involve tasks such as sentiment analysis, topic modeling, named entity recognition, and others. Text analysis and understanding systems are often used in applications such as medical text analysis, social media analysis, customer feedback analysis, and document summarization. These systems can help organizations gain insights from large amounts of unstructured text data and make data-driven decisions.

Text analysis and understanding is an important task in the answering of multimodal medical questions, as it involves extracting relevant information from the text and understanding the underlying meaning. There are various approaches to text analysis and understanding, including rule-based methods, which rely on pre-defined rules and patterns to extract information, and machine learning-based methods. In multimodal medical question answering, text analysis and understanding can be used to extract relevant information from medical literature, electronic health records, and other sources, and to generate natural language responses to questions.

NLP is a crucial component of multimodal medical question answering systems. It involves extracting relevant information from text data and understanding the underlying meaning. It can help organizations gain insights from large amounts of unstructured text data. With the advancement of NLP techniques, there is a great potential for the development of more accurate and efficient multimodal medical question answering systems that can assist healthcare professionals in making data-driven decisions.

*2.2. Image Analysis and Understanding*

Medical images are usually analyzed by radiologists and physicians who have been trained to recognize specific patterns. These patterns can indicate the presence of certain conditions or diseases. However, with the advancement in artificial intelligence, there is ongoing research to develop computer-aided diagnostic systems to assist human experts in interpreting medical images. The new algorithms and techniques mostly focus on automatically extracting information and understanding the visual content of images.

Medical imaging is a critical component of healthcare, as it enables healthcare professionals to diagnose and treat diseases and conditions accurately. Medical image analysis is a complex task that involves many steps, including image preprocessing, feature extraction, image segmentation, and classification. These steps can be time-consuming and challenging, especially when dealing with large datasets. However, automated medical image analysis systems that utilize machine learning algorithms can help streamline the process and improve accuracy. One of the significant advantages of using automated medical image analysis systems is their ability to detect subtle changes in medical images that may not be apparent to the human eye. These changes may indicate the presence of a disease or condition that may have been missed during manual analysis. By detecting these changes early, healthcare professionals can initiate treatment promptly, improving patient outcomes.

Medical image analysis can be used in various applications, such as detecting tumors, analyzing blood vessels, and monitoring disease progression. These applications can help healthcare professionals make better informed decisions about patient care, leading to better health outcomes. In multimodal medical question answering, automated medical image analysis can be integrated with natural language processing to generate comprehensive responses to questions about medical images. The integration of automated medical image analysis and natural language processing can provide a powerful tool for healthcare professionals to improve diagnosis and treatment planning.

*2.3. Video Analysis and Understanding*

Video analysis and understanding focuses on developing algorithms and techniques for automatically extracting information and understanding the content of videos. This can involve tasks such as object tracking, activity recognition, and scene understanding. Video analysis capability can be used in applications such as security and surveillance, medical informatics, sports analysis, and entertainment. Video analysis and understanding involves extracting relevant information from videos and understanding their underlying meaning. There are various approaches to video analysis and understanding, including manual methods, which involve a human expert manually analyzing the videos, and automated methods, which use machine learning algorithms to automatically analyze the videos.

In multimodal medical question answering, video analysis and understanding can be used to extract information from medical videos, such as surgical procedures or physical therapy sessions, and to generate natural language responses to video questions. Automated video analysis and understanding algorithms can help to speed up the process of analyzing and interpreting medical videos, and can be particularly useful in cases where there is a shortage of human experts available to manually analyze the videos.

Another application of video analysis and understanding in medicine is in medical education and training. Videos can be used to capture surgical procedures or other medical interventions, and these videos can be analyzed and annotated to provide feedback to medical students or residents. Video analysis and understanding can also be used to

simulate medical procedures and interventions, allowing medical professionals to practice and improve their skills in a safe and controlled environment.

Integration of multiple modalities, such as text, images, and videos, can further enhance the performance of multimodal medical question answering systems. By combining information from different modalities, these systems can provide more comprehensive and accurate responses to complex medical questions. For example, a question about a medical condition may require both textual information from medical literature and visual information from medical images or videos. The ability to seamlessly integrate and analyze information from multiple modalities is an important area of research in multimodal medical question answering. With the continued advancement in artificial intelligence and machine learning, multimodal medical question answering systems have the potential to revolutionize the way medical professionals diagnose, treat, and educate patients.

### 3. Datasets

In the medical domain there are various text, image and video based datasets that can be used for multimodal question answering research and development purposes. This study will briefly talk about text based medical question answering datasets and it will mainly focus on multimodal medical question answering with more emphasis on visual question answering datasets. Medical VQA datasets are listed In Table 1. Figure 1 illustrates an example of a medical video question answering system.

**Figure 1.** An example of a health-related question, textual answer, video containing the answer, and visual answer (temporal segment) from the video which is denoted by a red square [7].

MIMIC-III is a large, freely available dataset developed by the MIT Lab for Computational Physiology, containing deidentified health data associated with more than 40,000 critical care patients. It includes text and structured data, such as demographics, vital signs, medications, and lab test results [11].

eICU Collaborative Research Database dataset contains data from more than 200,000 intensive care unit stays, including electronic health records, vital signs, and laboratory test results [12].

Text-based experimental question answering system CHiQA is developed by the National Institute of Health (NIH) and it provides answers to consumer health related questions. CHiQA is a research project at the National Library of Medicine focusing on AI approaches to understanding health-related questions asked by the general public. CHiQA uses patient-oriented information from NIH to answer the questions [13].

CheXpert dataset contains chest X-ray images and associated labels for 14 diseases and conditions. It includes chest radiograph interpretations, text reports describing the images.

It consists of 14 common chest radiographic observations with 224,316 chest radiographs of 65,240 patients [14].

Medical Question Answering Dataset (MedQuAD) consists of 47,457 medical question-answer pairs created from 12 National Institute of Health (NIH) websites. It is text based and it has medical questions about treatment, diagnosis, and side effects related to drugs [15].

### 3.1. MedVidQA Dataset

Visual question answering (VQA) dataset contains open-ended natural language questions about images [6]. It contains answers to corresponding questions about images and it can be used for training machine learning and deep learning models for visual question answering tasks. The most recent VQA dataset from April 2017 contains 204,721 images, 1,105,904 questions, and 11,059,040 ground truth answers not specific to any domain [16]. This dataset inspired building various visual question answering datasets in the medical domain and it can be used for building multimodal medical question answering models. Medical visual question answering datasets are small compared with other visual question answering datasets such as DocVQA [17], MUST-VQA [18], and VQA [19].

Medical Video Question Answering dataset [20] is a collection of 3010 manually created health-related questions and timestamps as visual answers to those questions from trusted video sources, such as accredited medical schools with an established reputation, health institutes, health education, and medical practitioners. The MedVidQA dataset is developed as part of the medical video question answering challenge in BioNLP workshop at ACL 2022 [7]. The data preparation steps for MedVidQA dataset is illustrated in Figure 2.

### 3.2. VQA-Med-2021 Dataset

VQA-Med-2021 dataset is similar to VQA-Med-2020 and it consists of 4500 radiology images with 4500 question answer pairs for VQA data and 85 radiology images based on their visual content with 200 questions for VQG data. This edition focuses on answering questions about abnormalities in radiology images [21].

### 3.3. SLAKE Dataset

SLAKE, A Semantically-Labeled Knowledge-Enhanced Dataset for Medical Visual Question Answering is a bilingual dataset consisting of comprehensive semantic labels that are annotated by expert physicians. It consists of 642 images with 14,028 question-answer pairs in both English and Chinese. Questions are proposed by experienced doctors using a developed template annotation system [22].

### 3.4. VQA-Med-2020 Dataset

VQA-Med 2020 dataset is the third edition of the VQA-Med and it consists of 4000 radiology images with 4000 Question-Answer (QA) pairs as training data. This was the first time visual question generation (VQG) to generate natural language questions related to radiology image content was introduced [23].

### 3.5. PathVQA Dataset

PathVQA dataset consists of pathology images with captions that are extracted from online textbooks and digital libraries. It has 4998 pathology images with 32,799 open-ended questions that are manually checked for accuracy. The goal of the dataset was achieving an accuracy to the level to become a board-certified pathologist by passing the certification exam given by the American Board of Pathology (ABP) [24]. PyPDF2 and PDFMiner tools were used for extracting pathology images and captions using the Stanford CoreNLP toolkit from digital pathology textbooks and online digital libraries.

### 3.6. RadVisDial Dataset

RadVisDial dataset is a visual dialog in radiology, specific to chest X-ray images. It is derived from MIMIC-CXR [25] which is an openly available database of chest X-ray images. All the images that don't have the condition information were discarded when creating the dataset. As a result, the dataset contains 91,060 images. It has two datasets divided into silver-standard and gold-standard. The silver-standard dataset includes synthetically created dialogues and the gold-standard dataset includes natural dialog by two expert radiologists about a particular chest X-ray.

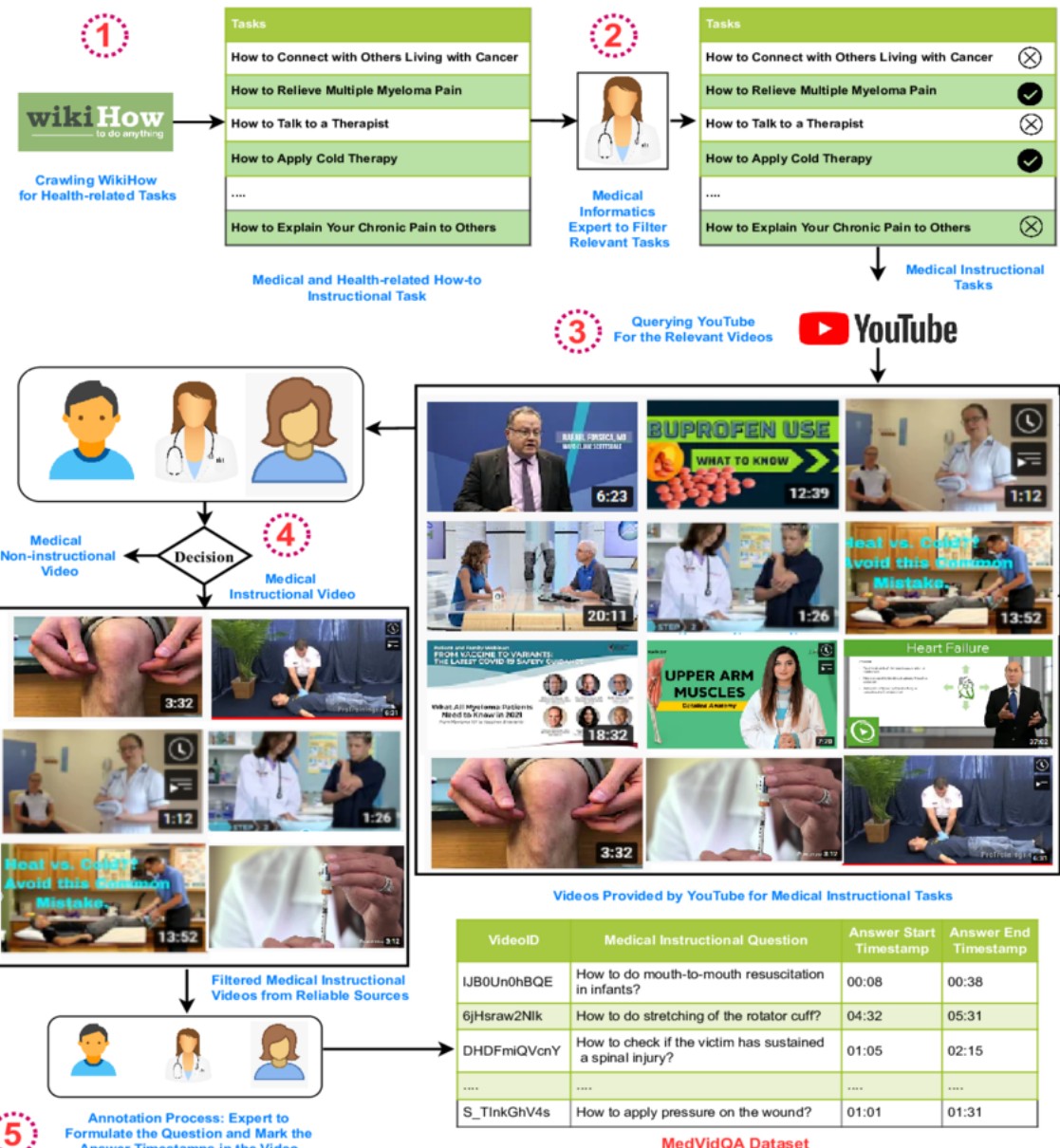

**Figure 2.** MedVidQA dataset creation steps, each step is numbered with circles in sequential order [20].

**Table 1.** Available Datasets For Multimodal Medical Visual Question Answering.

| Name | Article Link | Data Link | Dataset Size |
|---|---|---|---|
| **MedVidQA** [20] | https://arxiv.org/pdf/2201.12888.pdf (accessed on 27 December 2023) | https://doi.org/10.17605/OSF.IO/PC594 (accessed on 27 December 2023) | 6117 videos for MVC, 899 videos for the MVAL task. |
| **VQA-Med 2021** [21] | https://ceur-ws.org/Vol-2936/paper-87.pdf (accessed on 27 December 2023) | https://www.aicrowd.com/challenges/imageclef-2021-vqa-med-vqa (accessed on 27 December 2023) | 4500 radiology images |
| **SLAKE** [22] | https://arxiv.org/abs/2102.09542 (accessed on 27 December 2023) | https://www.med-vqa.com/slake/ (accessed on 27 December 2023) | 642 radiology images |
| **VQA-Med 2020** [26] | https://ceur-ws.org/Vol-2696/paper_106.pdf | https://www.aicrowd.com/challenges/imageclef-2020-vqa-med-vqg (accessed on 27 December 2023) | 4000 radiology images |
| **PathVQA** [24] | https://arxiv.org/abs/2003.10286 (accessed on 27 December 2023) | https://github.com/UCSD-AI4H/PathVQA (accessed on 27 December 2023) | 4998 pathology images |
| **RadVisDial** [26] | https://aclanthology.org/2020.bionlp-1.6.pdf (accessed on 27 December 2023) | https://physionet.org/content/mimic-cxr/ (accessed on 27 December 2023) | 91,060 radiology images |
| **VQA-Med 2019** [26] | https://ceur-ws.org/Vol-2380/paper_272.pdf (accessed on 27 December 2023) | https://www.aicrowd.com/challenges/imageclef-2019-vqa-med (accessed on 27 December 2023) | 4200 radiology images |
| **VQA-RAD** [27] | https://www.nature.com/articles/sdata2018251 (accessed on 27 December 2023) | https://osf.io/89kps/ (accessed on 27 December 2023) | 315 radiological images |
| **VQA-Med 2018** [28] | https://ceur-ws.org/Vol-2125/paper_212.pdf (accessed on 27 December 2023) | https://www.aicrowd.com/challenges/imageclef-2018-vqa-med (accessed on 27 December 2023) | 2866 medical images extracted from PubMed Articles |
| **Visual Genome** [29] | https://link.springer.com/article/10.1007/s11263-016-0981-7 (accessed on 27 December 2023) | https://https://visualgenome.org (accessed on 27 December 2023) | 101,174 images, 1.7 million questions |
| **OK-VQA** [30] | https://link.springer.com/article/10.1007/s11263-016-0981-7 (accessed on 27 December 2023) | https://https://https://okvqa.allenai.org (accessed on 27 December 2023) | 14,031 images, 14,055 questions |

### 3.7. VQA-Med-2019 Dataset

VQA-Med 2019 is a dataset containing 4200 radiology images and 15,292 question-answer pairs. It is an enhanced and larger version of the VQA-Med 2018 dataset and it was inspired from the VQA-RAD dataset. It contains CT/MRI images from the MedPix database and it targets the most frequent question categories with filters based on categories, modalities, planes, captions, localities, and diagnosis methods [26].

### 3.8. VQA-RAD Dataset

VQA-RAD is a dataset containing 315 corresponding radiological images for visual question answering tasks in the medical domain, specifically focusing on radiology [27]. The dataset contains radiology images from the MedPix database. It was manually created and validated by clinicians to provide a more specific and focused dataset for the VQA task in the medical domain, as radiology images have their own unique features and challenges compared with other types of medical image. The VQA-RAD dataset has been used to train and evaluate various deep learning models for the VQA task in the radiology domain.

### 3.9. VQA-Med-2018 Dataset

VQA-Med-2018 is a dataset for visual question answering in the medical domain. It contains medical images (such as X-rays and CT scans) paired with questions about the images, and the goal is for machine learning models to generate answers to the questions based on the image content [28]. It is designed to cover a wide range of medical conditions and modalities, and includes a diverse set of questions and answers to test the ability of VQA models to understand and generate answers based on medical images. The dataset has 6413 question-answer pairs and 2866 medical images extracted from PubMed Central articles.

### 3.10. Visual Genome Dataset

Visual Genome dataset serves as a bridge between language and vision. The dataset is created through crowdsourcing and provides rich annotations for images, enabling a deeper understanding of visual content [29]. The annotations encompass a comprehensive range of visual elements, including objects, attributes, relationships, and scene graphs. By connecting language descriptions with visual understanding, the Visual Genome dataset facilitates various computer vision tasks, such as object recognition, image captioning, and visual question answering. This paper details the data collection process, dataset statistics, and demonstrates the dataset's utility in advancing research in visual understanding and reasoning.

### 3.11. OK-VQA Dataset

OK-VQA dataset is a benchmark for visual question answering tasks that require external knowledge. Unlike traditional VQA datasets, OK-VQA includes questions that necessitate understanding and reasoning with external textual knowledge sources. The dataset focuses on complex questions that go beyond visual content and require access to external information for accurate answering.

## 4. Methods Used in Multimodal Medical Question Answering Research

Different methods have been employed for multimodal question answering by scientists. A search on Google Scholar was performed with the keywords "Multimodal Medical Question Answering" to find the methods used in the MMQA field. A variety of methods aim to effectively integrate information from multiple modalities, such as videos, images, and text. One common approach is the use of fusion-based methods, where features from different modalities are combined at different levels [31]. This can involve early fusion, where features are merged at the input level, or late fusion, where features are combined at a higher stage of processing. Another approach is attention-based methods, which selectively attend to relevant modalities or parts of the input [32]. Attention mechanisms allow the

model to focus on relevant visual and textual information, improving the accuracy of the answers. Additionally, graph-based methods have been employed to model the relationships between different modalities [33]. Graph structures can capture the dependencies and interactions between visual and textual elements, enabling more comprehensive understanding and reasoning [34]. Reinforcement learning techniques have also been applied, where the model learns to generate answers by interacting with the environment and receiving rewards. This approach allows for iterative refinement and improved performance over time. The choice of method depends on the specific requirements of the multimodal question answering task and the available data. Researchers continue to explore and develop new methods to enhance the performance and capabilities of multimodal question answering systems. In Table 2 different methods are listed and a couple of them are explained in the following paragraphs. Google researchers have proposed a vision-encoder text-decoder architecture for multimodal tasks called MaMMUT [35]. It is an interesting architecture in the sense that it performs joint training for vision language models.

*Performance Metrics of Existing Methods*

Evaluation metrics for multimodal medical QA include both text-based metrics, such as precision and recall, and visual-based metrics, such as image similarity. It will be best to use evaluation metrics which consider both textual and visual information to evaluate the performance of multimodal QA systems. The developed model will be evaluated using the metrics of the baseline model of MedVidQA to achieve a healthy comparison. The evaluation will also include a user study to assess the effectiveness and user satisfaction with the developed system.

The performance of medical visual question answering (VQA) systems is evaluated using two types of metrics: classification-based metrics and language-based metrics. Classification-based metrics, such as accuracy and F1 score, are commonly used to measure the system's performance by treating the answer as a classification result. These metrics calculate the exact match accuracy, precision, recall, and other relevant measures.

Language-based metrics, on the other hand, are borrowed from sentence evaluation tasks like translation and image captioning. In the medical VQA domain, tasks such as VQA-Med-2018, VQA-Med-2019, PathVQA, VQA-Med-2020, and VQA-Med-2021 employ language-based metrics. One commonly used language-based metric is BLEU [36]. Originally designed for machine translation, BLEU measures the similarity of phrases (n-grams) between two sentences. However, considering that the ground truth answers in medical VQA are typically shorter than those in machine translation or medical report generation tasks, the suitability of BLEU as a metric is questionable [37]. Additionally, in medical VQA, the semantic relevance of the answer is often more important than word-for-word matching. Nonetheless, BLEU could prove valuable when the answer corpus in medical VQA datasets becomes extensive and includes comprehensive sentences.

Apart from these general metrics, there are custom metrics specifically designed for medical VQA. For instance, VQA-Med-2018 introduced the Word-based Semantic Similarity (WBSS) and Concept-based Semantic Similarity (CBSS) metrics as alternatives [28]. These metrics aim to assess the semantic similarity between the answer and ground truth using word-level and concept-level representations [4]. It is worth noting that these custom metrics are not fixed components of the dataset or the approach, allowing researchers to choose more suitable metrics for evaluating their medical VQA systems [28]. Some researchers have even utilized the AUC-ROC (Area under the ROC Curve) metric as a classification-based evaluation measure to better assess the separability of answers [38].

**Table 2.** Available Methods for Multimodal Medical Visual Question Answering.

| Name | Article Link | Datasets |
|------|-------------|----------|
| **MedFuseNet** [38] | https://www.nature.com/articles/s41598-021-98390-1 (accessed on 27 December 2023) | MED-VQA, PathVQA |
| **MMBERT** [39] | https://arxiv.org/pdf/2104.01394.pdf (accessed on 27 December 2023) | VQA Med 2019, VQA-RAD |
| **QC-MLB** [16] | https://ieeexplore.ieee.org/abstract/document/9024133 (accessed on 27 December 2023) | VQA Med 2019 |
| **KEML** [40] | https://link.springer.com/chapter/10.1007/978-3-030-63820-7_22 (accessed on 27 December 2023) | VQA Med 2019 |
| **CGMVQA** [41] | https://ieeexplore.ieee.org/stamp/stamp.jsp?arnumber=9032109 (accessed on 27 December 2023) | VQA Med 2019 |
| **Hanlin** [42] | https://ieeexplore.ieee.org/stamp/stamp.jsp?arnumber=9032109 (accessed on 27 December 2023) | VQA Med 2019 |
| **Minhvu** [43] | https://ceur-ws.org/Vol-2380/paper_64.pdf (accessed on 27 December 2023) | VQA Med 2019 |
| **TUA1** [44] | https://ceur-ws.org/Vol-2380/paper_190.pdf (accessed on 27 December 2023) | VQA Med 2019 |
| **QCMLB** [45] | https://ieeexplore.ieee.org/document/9024133 (accessed on 27 December 2023) | VQA Med 2019 |
| **UMMS** [46] | https://ceur-ws.org/Vol-2380/paper_123.pdf (accessed on 27 December 2023) | VQA Med 2019 |
| **IBM Research AI** [47] | https://ceur-ws.org/Vol-2380/paper_112.pdf (accessed on 27 December 2023) | VQA Med 2019 |
| **LIST** [48] | https://ceur-ws.org/Vol-2380/paper_124.pdf (accessed on 27 December 2023) | VQA Med 2019 |
| **TURNER.JCE** [49] | https://ceur-ws.org/Vol-2380/paper_116.pdf (accessed on 27 December 2023) | VQA Med 2019 |
| **JUST19** [50] | https://ceur-ws.org/Vol-2380/paper_125.pdf (accessed on 27 December 2023) | VQA Med 2019 |
| **Team_PwC_Med** [51] | https://ceur-ws.org/Vol-2380/paper_147.pdf (accessed on 27 December 2023) | VQA Med 2019 |
| **Techno** [52] | https://ceur-ws.org/Vol-2380/paper_117.pdf (accessed on 27 December 2023) | VQA Med 2019 |
| **Gasmi** [53] | https://www.tandfonline.com/doi/full/10.1080/01969722.2021.2018543 (accessed on 27 December 2023) | VQA Med 2019 |
| **Xception-GRU** [54] | https://ceur-ws.org/Vol-2380/paper_127.pdf (accessed on 27 December 2023) | VQA Med 2019 |
| **Thanki** [55] | https://ceur-ws.org/Vol-2380/paper_167.pdf (accessed on 27 December 2023) | VQA Med 2019 |
| **Chakri** [56] | https://ieeexplore.ieee.org/document/8987108 (accessed on 27 December 2023) | VQA Med 2018 |
| **UMMS** [46] | https://ceur-ws.org/Vol-2125/paper_212.pdf (accessed on 27 December 2023) | VQA Med 2018 |
| **TU** [57] | https://ceur-ws.org/Vol-2125/paper_107.pdf (accessed on 27 December 2023) | VQA Med 2018 |
| **HQS** [34] | https://www.sciencedirect.com/science/article/abs/pii/S0957417420307697?via%3Dihub (accessed on 27 December 2023) | VQA Med 2018 |
| **MMQ-VQA** [58] | https://link.springer.com/chapter/10.1007/978-3-030-87240-3_7 (accessed on 27 December 2023) | PathVQA, VQA-RAD |

**Table 2.** *Cont.*

| Name | Article Link | Datasets |
|---|---|---|
| **NLM** [59] | https://ceur-ws.org/Vol-2125/paper_165.pdf (accessed on 27 December 2023) | VQA Med 2018 |
| **JUST** [60] | https://ceur-ws.org/Vol-2125/paper_171.pdf (accessed on 27 December 2023) | VQA Med 2018 |
| **FSTT** [61] | https://ceur-ws.org/Vol-2125/paper_159.pdf (accessed on 27 December 2023) | VQA Med 2018 |
| **AIML** [62] | https://ceur-ws.org/Vol-2696/paper_78.pdf (accessed on 27 December 2023) | VQA Med 2020 |
| **TheInceptionTeam** [63] | https://ceur-ws.org/Vol-2696/paper_69.pdf (accessed on 27 December 2023) | VQA Med 2020 |
| **Bumjun-jung** [64] | https://ceur-ws.org/Vol-2696/paper_87.pdf (accessed on 27 December 2023) | VQA Med 2020 |
| **HCP-MIC** [65] | https://ceur-ws.org/Vol-2696/paper_74.pdf (accessed on 27 December 2023) | VQA Med 2020 |
| **NLM** [66] | https://ceur-ws.org/Vol-2696/paper_98.pdf (accessed on 27 December 2023) | VQA Med 2020 |
| **HARENDRAKV** [67] | https://ceur-ws.org/Vol-2696/paper_62.pdf (accessed on 27 December 2023) | VQA MEd 2020 |
| **Shengyan** [68] | https://ceur-ws.org/Vol-2696/paper_73.pdf (accessed on 27 December 2023) | VQA Med 2020 |
| **Kdevqa** [69] | https://ceur-ws.org/Vol-2696/paper_81.pdf (accessed on 27 December 2023) | VQA Med 2020 |
| **SYSU-HCP** [70] | https://ceur-ws.org/Vol-2936/paper-99.pdf (accessed on 27 December 2023) | VQA Med 2021 |
| **Yunnan** [71] | https://ceur-ws.org/Vol-2936/paper-120.pdf (accessed on 27 December 2023) | VQA Med 2021 |
| **TeamS** [72] | https://ceur-ws.org/Vol-2936/paper-98.pdf (accessed on 27 December 2023) | VQA MEd 2021 |
| **Lijie** [73] | https://ceur-ws.org/Vol-2936/paper-104.pdf (accessed on 27 December 2023) | VQA Med 2021 |
| **IALab_PUC** [74] | https://ceur-ws.org/Vol-2936/paper-113.pdf (accessed on 27 December 2023) | VQA MEd 2021 |
| **TAM** [75] | https://ceur-ws.org/Vol-2936/paper-106.pdf (accessed on 27 December 2023) | VQA Med 2021 |
| **Sheerin** [76] | https://ceur-ws.org/Vol-2936/paper-110.pdf (accessed on 27 December 2023) | VQA Med 2021 |
| **CMSA** [77] | https://dl.acm.org/doi/10.1145/3460426.3463584 (accessed on 27 December 2023) | VQA-RAD |
| **QCR, TCR** [78] | https://www4.comp.polyu.edu.hk/~csxmwu/papers/MM-2020-Med-VQA.pdf (accessed on 27 December 2023) | VQA-RAD |
| **MMQ** [79] | https://arxiv.org/abs/2105.08913 (accessed on 27 December 2023) | VQA-RAD |
| **MEVF** [80] | https://research.monash.edu/en/publications/overcoming-data-limitation-in-medical-visual-question-answering (accessed on 27 December 2023) | VQA-RAD |
| **CPRD** [81] | https://miccai2021.org/openaccess/paperlinks/2021/09/01/113-Paper1235.html (accessed on 27 December 2023) | VQA-RAD, SLAKE |
| **Silva** [82] | https://www.sciencedirect.com/science/article/pii/S2667305323000467 (accessed on 27 December 2023) | VQA-Med |

MedFuseNet method is an attention-based multimodal deep learning model for visual question answering in the medical domain [38]. It is designed to process and understand both the visual information contained in medical images and the natural language text of questions about those images. It uses an attention mechanism to dynamically weigh the contribution of different features from the image and text inputs, allowing it to effectively focus on the most relevant information for a given question. The model is trained on a large corpus of medical data, allowing it to learn rich representations of medical concepts and to generate accurate and relevant answers to a wide range of medical VQA tasks. The goal of MedFuseNet is to provide a powerful and flexible deep learning solution for medical VQA, enabling healthcare providers and researchers to quickly and easily gain insights from medical images and associated text data.

MMBERT, Multimodal BERT Pretraining for improved Medical VQA method is a pretrained language model that combines natural language processing (NLP) and computer vision techniques for medical visual question answering (VQA) [39]. The model is based on the popular BERT architecture and is trained on a large corpus of medical data that includes both text and images. The goal of MMBERT is to improve the performance of medical VQA tasks by leveraging the rich representations learned from multimodal data. By combining information from text and image inputs, MMBERT can better understand complex medical concepts and provide more accurate answers to questions about medical images.

QC-MLB, Question-Centric Model for Visual Question Answering in Medical Imaging method is a machine learning model for visual question answering (VQA) in medical imaging [45]. It is designed to be question-centric, meaning it focuses on the specific question being asked to generate an answer, rather than relying solely on image-level features or information. The model leverages transfer learning from pre-trained models in natural language processing and computer vision to extract features from both the question and the medical image. These features are then combined and processed by the QC-MLB model to generate an answer. The goal of QC-MLB is to provide accurate and reliable answers to questions about medical images in a fast and efficient manner, making it useful for clinical decision support and other applications in the field of medical imaging.

Li et al. compares their architecture VPTSL with others in the Figure 3 [83]. Overall, these methods (TAGV, VSLNet, VSLNet-L, ACRM, and RaNet) introduce innovative strategies to enhance various aspects of video question answering. They leverage query-guided highlighting, multi-scale approaches, attention mechanisms, and relation-aware networks to improve the accuracy and effectiveness of moment localization, temporal grounding, and cross-modal interactions in the context of video analysis and comprehension. They use MedVidQA dataset. The VPTSL method aims to bridge gaps by using subtitles and visual highlights to locate relevant text spans within the input question. They use pre-trained DeBerTa model. By integrating visual features into a pre-trained language model, they enhance the joint semantic representations. Through cross-modal interaction and video-text highlighting, they obtain highlight features aligned with the visual prompt. To address semantic differences, they design a text span predictor that encodes the question, subtitles, and prompted visual highlights. The formulated TAGV task predicts subtitle spans corresponding to the visual answer, facilitating a more comprehensive multimodal question answering approach.

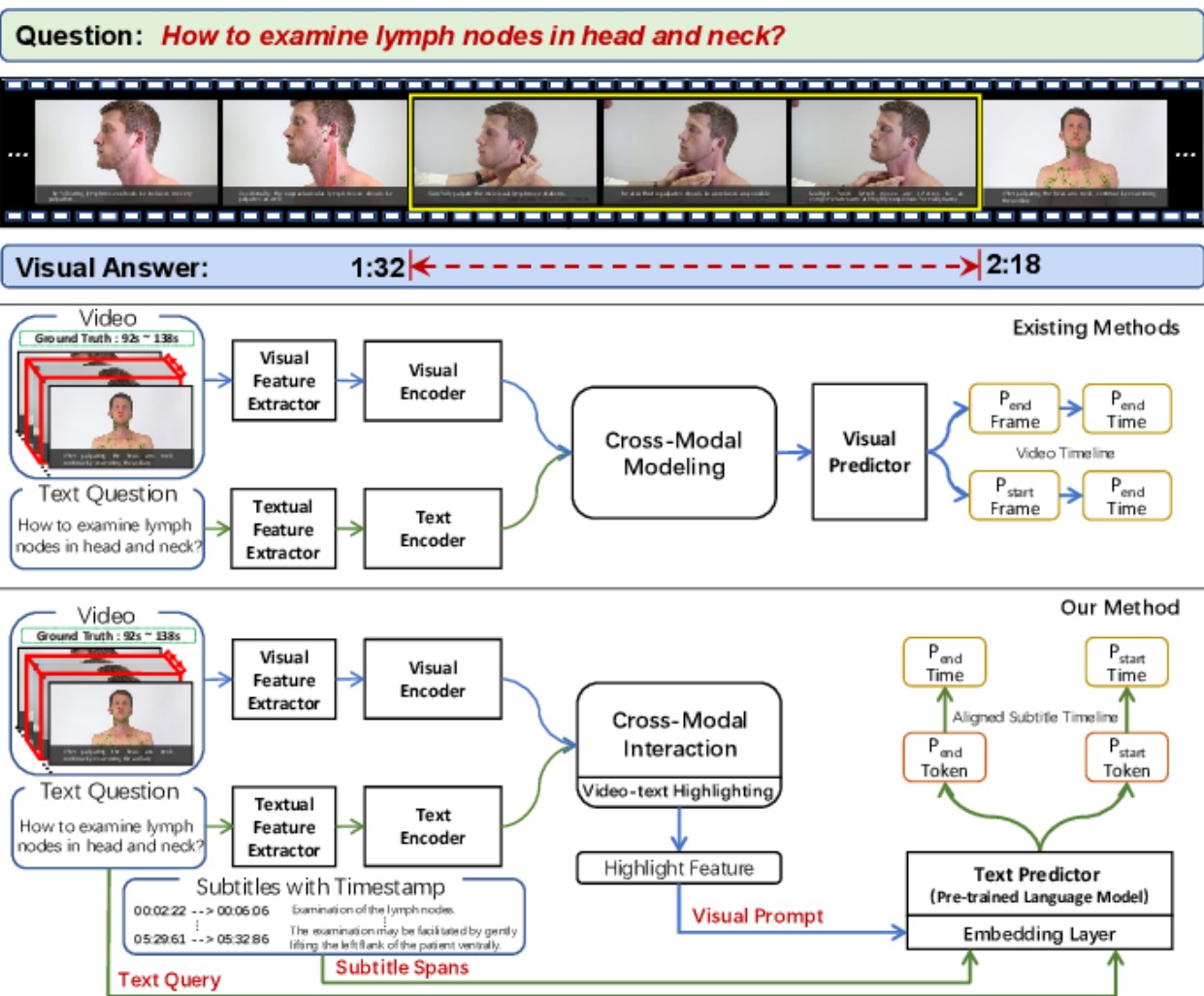

**Figure 3.** Comparison of architectures of medical multimodal question answering [83]. Yellow square shows the time span of the video containing the answer to the question.

TMLGA [84] model uses a dynamic filter which transfers language data to a visual domain attention map. They define a new loss function to choose the most relevant part of the video. Soft labels are used for annotation uncertainties.

VSLBase [85] is a span-based QA framework. It treats visual features like text passages, with the target moment as the answer span. It predicts start and end times of the visual answer span during training.

VSLNet [85] introduces Query-Guided Highlighting (QGH) to enhance the VSLBase model. It treats the target moment and adjacent contexts as foreground, while the rest is considered background. The foreground covers a slightly longer span than the answer span, providing more contextual information.

VSLNet-L [86] addresses the performance degradation on long videos by incorporating concepts from multi-paragraph question answering. It employs a multi-scale split-and-concatenation strategy to segment long videos into shorter clips. The hierarchical searching strategy is designed to improve moment localization accuracy, ensuring precise identification of relevant moments within the video.

ACRM [87] predicts temporal grounding by modeling the interaction between vision and language. It utilizes an attention module to assign hidden features to the query text, incorporating richer semantic information. This semantic information is crucial for effectively locating relevant video content. Additionally, ACRM employs an additional predictor that utilizes internal frames during training, resulting in improved localization accuracy.

RaNet [88] represents a relation-aware network inspired by reading comprehension. It formulates temporal language grounding in videos. The framework of RaNet selects a grounding moment from predefined answer collections using coarse-and-fine choice-query interaction and choice-choice relation construction. The choice-query interactor matches visual and textual information simultaneously at both the sentence-moment and token-moment levels, enabling a coarse-and-fine cross-modal interaction.

## 5. Large Language Models for Medical Question Answering

Large language models have changed the research landscape. The workflow that has been used in AI research has changed fundamentally [89]. Research done by pioneers of the AI field Andrew Ng, Yann Lecun, Geoff Hinton, Yoshua Bengio, Christopher Manning, Richard Socher, and Tomas Mikolov is also transforming the field [90–94].

Large language models have recently been used in medical question answering [95–97]. Large language models, such as GPT-3, are characterized by their massive scale, pre-training on vast amounts of text data, transfer learning for adapting to specific tasks, contextual understanding for capturing meaning based on context, language generation capabilities, natural language understanding to comprehend and interpret text, and fine-grained control over text generation. These models provide powerful and versatile tools for various natural language processing tasks, although they have limitations and considerations to be aware of, such as potential biases and the need for critical evaluation of their outputs.

We have tested ChatGPT's response to the challenge question. ChatGPT was very close to the timestamps challenge was looking for. The write-up showing the conversation with ChatGPT is attached to the end of the document. Researchers from medical fields have also explored the capabilities of ChatGPT. Ayers et al. have compared physician's responses to ChatGPT's responses to questions randomly selected from a social media forum and they concluded that ChatGPT gave more accurate responses [98]. Duong and Solomon [99], Oh et al. [100], and Antaki et al. [101] assessed GPT-3's performance in different medical fields such as genetics, surgery, and ophthalmology, respectively.

Moreover, Microsoft has released a product called BioGPT that answers medical questions [102]. A sample execution of BioGPT is also attached to the end of the document. There is not a web interface to talk to BioGPT; however, it can be embedded in the code, as illustrated in the attached pdf. Microsoft search tool Bing has a Health search page too. When searched, it can give videos as an answer. For instance, when searching "how to ease neck pain", videos were shown too. But, according to the use cases I have demonstrated, it does not go beyond being a health search engine.

Limitations and challenges of large language models for question answering are related to safety, bias, privacy, ethics, and correctness. Wrong answers can have serious consequences for human health. Also, language models, both ChatGPT and BioGPT, provide incorrect citations and can make false claims, which can be fatal when taken as a reference in health. Weng et al. addressed that issue in their papers and have discussed that large language models need in-depth thought for high-quality medical answers [103].

One of the large language model-supported medical question answering models is Med-Palm [104] by Google. An example of a MedPalm question answering model is seen in the Figure 4. It is mentioned that Flan-Palm developed by Google could not be used for medical question answering due to its limitations in terms of bias and scientific grounding. Google improved Flan-Palm for medical question answering and renamed the new product Med-Palm.

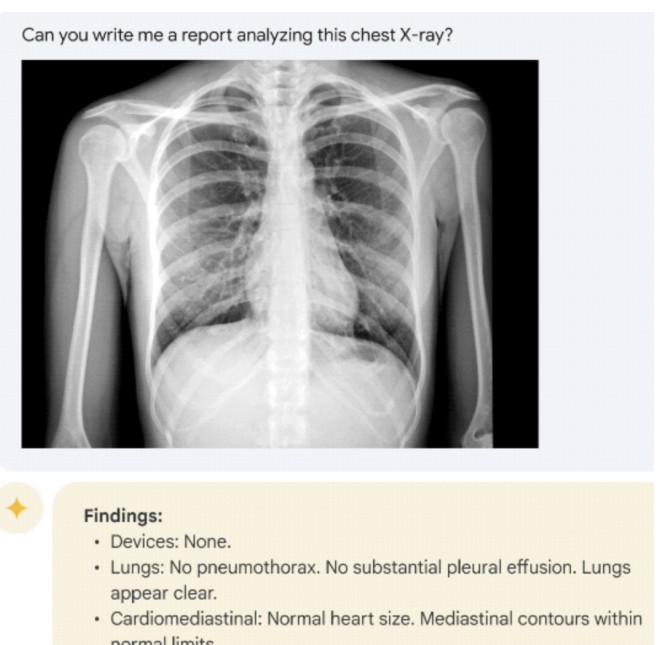

**Figure 4.** An example of a multimodal language model for the medical domain by Google [104].

There are several commercial providers of medical question answering services. These providers typically offer a platform or interface through which users can ask medical questions and receive answers from trained medical professionals or AI-powered systems. Some of these commercial platforms are listed in Table 3.

**Table 3.** Available Tools for Medical Question Answering.

| Tool Name | Tool Link |
| --- | --- |
| **AdaHealth** [105] | https://ada.com (accessed on 27 December 2023) |
| **Doctor on Demand** [106] | https://doctorondemand.com (accessed on 27 December 2023) |
| **WebMD** [107] | https://symptoms.webmd.com/ (accessed on 27 December 2023) |
| **Mayo Clinic** [108] | https://www.mayoclinic.org/symptoms (accessed on 27 December 2023) |
| **Google Health** [109] | https://health.google (accessed on 27 December 2023) |
| **Apple Health** [110] | https://www.apple.com/ios/health/ (accessed on 27 December 2023) |

## 6. Tools and Services

Ada Health is a digital health company that offers a mobile app and online platform for users to ask medical questions and receive personalized health recommendations based on their symptoms and medical history [105]. The Ada app is designed to help people better understand and manage their health by providing personalized health information and recommendations. Users can enter their symptoms, medical history, and other relevant information into the app, and the app will provide recommendations for self-care or further medical evaluation based on the user's specific circumstances, as seen in Figure 5. The Ada app is not intended to replace the advice of a healthcare professional, but rather to provide additional information and support to help users make informed decisions about their health.

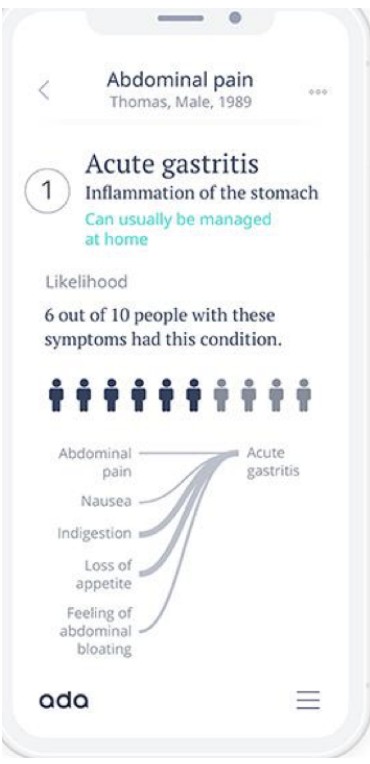

**Figure 5.** An example interface of the Ada health application [105].

Doctor On Demand is a telemedicine service that allows users to connect with doctors, therapists, and other healthcare providers through virtual video consultations. Users can ask medical questions and receive guidance on their health and wellness [106]. A sample user interface of the chat service it is providing to its patients can be see in Figure 6. The service is available through a mobile app and website, and users can access a range of healthcare services, including primary care, behavioral health, and specialty care. Doctor On Demand is intended to provide users with convenient, accessible, and affordable healthcare services, especially for those who may have difficulty getting to a physical healthcare facility.

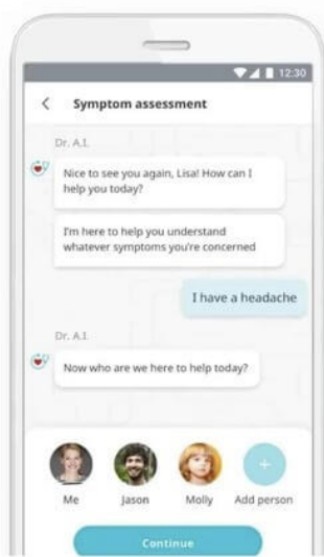

**Figure 6.** An example interface of the Doctor On Demand application [106].

WebMD is a digital health company that offers a variety of health and wellness resources, including a symptom checker and medical reference materials [107]. Users can ask medical questions and receive answers from trained medical professionals or AI-powered systems. The website and app offer a range of content, including news articles, medical reference materials, and tools for managing personal health. WebMD's Symptom Checker tool allows users to enter their symptoms and receive information about potential causes. Users can enter their symptoms and any relevant information, such as their age and gender. The tool then provides a list of possible conditions that could be causing the symptoms, along with recommended next steps.

Mayo Clinic is a nonprofit academic medical center that offers a range of health and wellness resources, including medical reference materials [108]. The website includes information on conditions, treatments, and procedures, as well as health and wellness tips. The Mayo Clinic website and app also offer a symptom checker, a medication tracker, and other tools to help users better understand and manage their health.

There are several large tech companies that offer remote health services, which allow users to ask health-related questions using a variety of methods, such as typing, speaking, or taking a photo. Google offers a service called Google Health, which allows users to ask health-related questions and provides information and resources on a wide range of health topics [109]. Apple has a similar service called Apple Health, which provides information and tools for tracking and managing various aspects of users' health [110].

### 6.1. Clinical Research

There is a growing demand for data scientists and analysts with expertise in healthcare data. Supporting research in this area could involve developing new tools and technologies for analyzing and utilizing healthcare data. Research in the medical field can improve the quality of healthcare and help address various health challenges. The development of telehealth and remote monitoring technology such as chatbots or virtual assistants that can answer medical questions can improve access to healthcare.

Clinical trials are a key part of the process of developing new treatments and therapies. Researchers may use multimodal medical question answering in clinical trials in order to test the safety and effectiveness of new treatments. It is possible to extract information from medical records and clinical trial data to identify patterns and biomarkers that can be used to predict treatment response.

### 6.2. Investment

There are many potential angles from which to approach the topic of investment in the Multimodal Medical Question Answering field. The demand for development of Multimodal Medical Question Answering systems is an active area of potential investment, with the goal of creating systems that can help improve the accuracy and efficiency of medical diagnosis and treatment by providing doctors and other healthcare professionals with more comprehensive and accurate information about their patients' medical conditions. Applying artificial intelligence to medical technologies such as electronic health records and telemedicine can improve the efficiency of healthcare delivery, helping providers see more patients and reducing wait times.

New technological infrastructure investments in the healthcare industry are becoming increasingly popular to assist in providing accurate diagnoses and treatment plans. Investment in this area can include funding for research and development, as well as the implementation and integration of these systems into clinical settings. Many medical companies and organizations invest in research and development in order to discover new treatments and technologies, and this can lead to reducing the overall cost of healthcare by improving efficiency and reducing the need for costly procedures.

Startups

There are several startup companies that offer medical question answering services. These startups typically use AI and machine learning technologies to provide users with personalized health recommendations based on their symptoms and medical history.

Buoy Health offers a symptom checker and triage tool that helps users understand their health concerns and connect with the appropriate level of care [111].

K Health offers a symptom checker and personalized health guidance platform that uses AI and machine learning to provide users with recommendations based on their symptoms and medical history [112].

Medwise.ai finds relevant answers to clinical questions. It is a question-answering search tool and it is intended to support healthcare professionals such as doctors, nurses, and other medical professionals at the point of care[113].

## 7. Challenges of the Research Field

Medical visual question answering (VQA) research involves the intersection of medical imaging, natural language processing (NLP), and computer vision. While significant progress has been made in this field, there are several challenges and open areas that researchers are actively working on. Here are some of them:

- Limited annotated medical data. Annotated medical imaging datasets for VQA are relatively small compared with general visual question answering datasets;
- Domain-specific knowledge representation. Medical VQA requires understanding complex medical concepts, terminology, and anatomical structures;
- Interpretable and explainable answers. In medical applications, interpretability and explainability of the VQA models are critical;
- Handling variability and uncertainty. Medical imaging exhibits significant variability due to variations in patient demographics, imaging techniques, and pathology;
- Multimodal fusion and alignment. Medical VQA requires effectively fusing information from both the visual and textual modalities;
- Generalization across tasks and domains. Medical VQA models trained on one medical imaging modality or clinical task may not generalize well to other modalities or tasks;
- Ethical considerations and bias. As with any AI application in healthcare, addressing ethical considerations and mitigating biases is important in medical VQA;
- Real-time performance. In clinical settings, real-time or near-real-time performance is often required for VQA systems.

## 8. Summary and Conclusions

This research presented a literature review of the current state-of-the-art techniques for multimodal medical question answering and discussed the importance of developing such systems to improve healthcare decision-making. The introduction highlighted the need for efficient clinical decision support systems in healthcare. Section 2 reviewed prior research on multimodal medical question answering using natural language processing and computer vision techniques. Section 3 focused on the available multimodal QA datasets. It was apparent in Section 3 that there are not many QA golden datasets as it is not straightforward to answer medical questions. Section 4, the methods section, outlined the research frameworks, algorithms, and evaluation metrics used for testing multimodal medical question answering systems. Section 5 explored state-of-the-art large language models in healthcare QA and it found that large language models have changed the QA research landscape fundamentally. Sections 6 and 7 presented methodologies and preliminary outcomes, with a touch on industrial research and startup investment opportunities.

In conclusion, efficient and accurate clinical decision support systems in healthcare are essential. The development of a multimodal medical question answering system that incorporates text, images, and videos can improve the accuracy and speed of answering complex medical questions. The findings of this research can contribute to the advancement

of healthcare decision support systems by providing a more comprehensive and efficient approach to clinical decision-making.

**Author Contributions:** H.D. performed most of the analysis and the writing. W.Z. helped set up the direction of research and contributed to the writing of the paper. All authors have read and agreed to the published version of the manuscript.

**Funding:** During the writing of this article, the second author (WZ) was partly funded by the National Science Foundation (NSF) grant number 2141124.

**Institutional Review Board Statement:** Not applicable.

**Informed Consent Statement:** Not applicable.

**Data Availability Statement:** Data sharing not applicable.

**Conflicts of Interest:** The authors declare no conflicts of interest.

**Abbreviations**

The following abbreviations are used in this manuscript:

| | |
|---|---|
| AI | Artificial Intelligence |
| NLP | Natural Language Processing |
| VQA | Visual Question Answering |
| MMQA | Multimodal Medical Question Answering |
| MVQA | Medical Visual Question Answering |
| AI | Artificial Intelligence |
| FCM | Fuzzy Cognitive Maps |
| BioASQ | Biomedical Semantic Indexing and Question Answering |
| MedVidQA | Medical Video Question Answering Challenge |
| BERT | Bidirectional Encoder Representations from Transformers |
| ABP | American Board of Pathology |

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
