# Peer review of "Survey of Multimodal Medical Question Answering"

_biomedinformatics, doi:10.3390/biomedinformatics4010004_

Round 1

Reviewer 1 Report

Comments and Suggestions for Authors

Abstract

1. The keywords should be aligned with MESh terminology (https://www.ncbi.nlm.nih.gov/mesh/)

Introduction

2. It would be advisable to provide a more detailed explanation of the reasons behind the necessity of multimodal question answering systems. What has driven the increased interest in this field in recent years?

3. In the introduction section, the authors should clearly state the research objectives.

4. What is the purpose of describing the article's structure in the introduction section?

Materials and Methods

1. Why is the literature search source mentioned in the abstract not included in the Methods section?

2. Why does the research objective in the abstract not correspond to the research objective in the Methods section?

3. How was the literature search conducted in the Google Scholars database?

4. How many publications were selected in total for this study?

5. What criteria were used for the selection of publications?

6. How many individuals were involved in the publication selection process?

Reviewer 2 Report

Comments and Suggestions for Authors

Authors have provided a literature review on multimodal medical question answering.

Overall, the English language quality is good, and the content is informative.

Some sentences are quite lengthy and could be broken down into shorter, more digestible sentences for easier reading.

Focus on providing essential information and avoid excessive repetition.

Please before using abbreviations define them.

Screenshots of user interfaces of tools and services you mentioned, like Ada Health or Doctor On Demand, can provide a visual reference for readers.

There is a missed reference on the page 12.

Round 2

Reviewer 1 Report

Comments and Suggestions for Authors

No more comment

Reviewer 2 Report

Comments and Suggestions for Authors

Thanks for considering my comments.